# Stannate-Based Materials as Anodes in Lithium-Ion and Sodium-Ion Batteries: A Review

**DOI:** 10.3390/molecules28135037

**Published:** 2023-06-27

**Authors:** You-Kang Duan, Zhi-Wei Li, Shi-Chun Zhang, Tong Su, Zhi-Hong Zhang, Ai-Jun Jiao, Zhen-Hai Fu

**Affiliations:** 1Key Laboratory of Comprehensive and Highly Efficient Utilization of Salt Lake Resources, Qinghai Institute of Salt Lakes, Chinese Academy of Sciences, Xining 810008, China; 2Key Laboratory of Salt Lake Resources Chemistry of Qinghai Province, Xining 810008, China; 3University of Chinese Academy of Sciences, Beijing 100049, China

**Keywords:** stannate-based anodes, composition and structure design, energy storage mechanism, electrochemical performances

## Abstract

Binary metal oxide stannate (M_2_SnO_4_; M = Zn, Mn, Co, etc.) structures, with their high theoretical capacity, superior lithium storage mechanism and suitable operating voltage, as well as their dual suitability for lithium-ion batteries (LIBs) and sodium-ion batteries (SIBs), are strong candidates for next-generation anode materials. However, the capacity deterioration caused by the severe volume expansion problem during the insertion/extraction of lithium or sodium ions during cycling of M_2_SnO_4_-based anode materials is difficult to avoid, which greatly affects their practical applications. Strategies often employed by researchers to address this problem include nanosizing the material size, designing suitable structures, doping with carbon materials and heteroatoms, metal–organic framework (MOF) derivation and constructing heterostructures. In this paper, the advantages and issues of M_2_SnO_4_-based materials are analyzed, and the strategies to solve the issues are discussed in order to promote the theoretical work and practical application of M_2_SnO_4_-based anode materials.

## 1. Introduction

With the rapid development of human society in recent years, the need for energy has grown significantly. Environmental pollution caused by greenhouse gases such as carbon dioxide produced during the consumption of traditional fossil energy has prompted people to accelerate the development and use of green and clean energy [1,2,3]. However, renewable energy sources such as solar and wind are intermittent and diverse, and their large-scale expansion requires efficient and stable energy storage equipment [4,5,6]. Of all energy storage devices, alkali metal ion batteries are considered to be the most promising because of their high energy density, extended cycle life and environmental friendliness [7,8,9]. Since their introduction, lithium-ion batteries (LIBs) have become a mainstream energy storage device for mobile electronic devices, electric vehicles and other equipment [7,10]. However, commercial LIBs currently available are still unable to meet the requirements of large-scale energy storage devices such as electric vehicles and smart grids in terms of size, power output and energy density [11]. In the case of electric vehicles, there is diversity in the choice of cathode materials. BMW uses lithium nickel manganese cobalt oxide (NMC), while Tesla uses lithium nickel cobalt aluminum oxide (NCA), and electric vehicle companies in China often use lithium iron phosphate (LFP). Currently, carbonaceous materials are generally chosen as anode materials [12]. However, carbonaceous materials have a lower theoretical capacity (372 mAh·g^−1^) and take up about half the volume of the battery. Hence, higher-capacity anode materials need to be developed to replace carbon materials to match the high capacity of the anode materials and improve the energy storage capacity of LIBs. Nevertheless, the continued development and application of LIBs are leading to the rapid depletion of lithium resources, yet sodium resources are quite abundant. Therefore, SIBs, which use sodium resources as the main raw material, can be developed to replace a portion of LIBs, contributing to cost reduction and sustainability [13,14]. SIBs have a similar electrochemical storage mechanism to that of LIBs and can be adapted from LIB technology, allowing them to develop rapidly. Due to their significant advantage of lower cost, SIBs are likely to replace several LIBs as an important classification of energy storage devices. However, due to the relatively large radius of Na^+^ (0.59 Å for Li^+^ and 1.02 Å for Na^+^ in radius), significant volume expansion problems occur during the sodiation/desodiation process, resulting in poor energy density and cycle stability [15,16,17]. Due to this issue, Na^+^ is difficult to insert between the lattice layers of graphite; therefore, graphite anode material commonly used in LIBs cannot be applied to SIBs. Thus, selecting the right anode material is a key issue for both LIBs and SIBs.

Among the currently studied anode materials, which mainly include graphite, tin, germanium and silicon, a comparison of the capacity and volume variation of the elements in the IVA group is shown (Figure 1). Although the theoretical specific capacity of Sn is not the highest of them all, the volumetric specific capacity of Sn is quite close to that of Si and Ge. In particular, Sn has a much higher electrical conductivity (9.17 × 10^4^ S cm^−1^) than several other materials. Thus, Sn-based materials such as Sn, SnO_2_, SnS, M_2_SnO_4_ and their composites have been extensively researched as they can have a very high theoretical capacity and a suitable potential platform [18,19,20,21]. Due to conversion and alloying reactions, the theoretical capacity of low-cost Sn dioxide can reach 1494 mAh·g^−1^, with the conversion reaction contributing 711 mAh·g^−1^ and the alloying reaction contributing 783 mAh·g^−1^. This is much higher than that of commonly used graphite materials, making Sn-based oxides a potential candidate for anode electrode materials [22,23,24].

As researchers investigated further, several serious problems with the Sn-based material were identified: (i) During the insertion/extraction of Li^+^ and Na^+^, Sn fully converts to Li_22_Sn_5_ and Na_15_Sn_4_, undergoing severe volume expansion (260% and 420%, respectively). This causes pulverization of the active material, allowing it to break away from the collector and leading to electrode failure [25,26,27,28]. This is detrimental to the cycling stability and reversible capacity of the battery. (ii) The inherent poor electrical and ionic conductivity of the material results in slow reaction kinetics, which reduces high-magnification performance. (iii) The poor reversibility of the conversion reaction between Sn and Li_2_O or Na_2_O during the initial cycle results in a large initial irreversible capacity loss as well as a low initial coulombic efficiency (ICE) [29]. In response to these problems, several studies have shown that the doping of Sn-based materials with other transition metal elements to produce metal oxide M_2_SnO_4_ can improve the properties [30,31,32]. On the one hand, due to the differences in expansion coefficients between different metal elements, redox reactions are enriched during the ion insertion/extraction process. This forms metal “M” or metal oxide “MO_y_” with different redox potentials, which can act as buffer substrates between each other to slow down the volume changes during cycling, thus ensuring the structural stability of the electrode material and reducing the capacity decay [33,34,35]. Furthermore, the conversion reaction of M can provide Sn with the required Li_2_O or Na_2_O, promoting the reversibility of the Sn conversion reaction. This improves the poor reversibility of the conversion reaction and reduces irreversible capacity loss [36]. On the other hand, combining different nanocrystals allows for the construction of heterogeneous structures at the contact interface. The internal electric field formed at the interface improves surface reaction kinetics, facilitating charge transfer and increasing material conductivity. For example, Mn_2_SnO_4_, a combination of Sn and Mn, has “metal-like conductivity” and improves the reaction kinetics [34,37]. In addition to overcoming the problems of tin-based materials, researchers have combined other modification strategies such as nanosizing, carbon compounding, heterostructuring and heteroatom doping.In recent years, stannate-based materials have gained great attention due to their superiority as anodes in LIBs and SIBs. In order to make stannate-based materials a viable alternative to anode materials, researchers have conducted a lot of exploration and proposed various modification strategies to address their problems. Table 1 provides a review of articles from the last decade on the research of different Sn-based materials. Among these excellent reviews, the main materials targeted include metallic Sn, SnO, SnO_2_, SnS, M_2_SnO_4_ and their composites, as shown in Table 1. For example, Huang Bin et al. [38] summarized studies on the mechanisms of action and institutional design of Sn-based oxides, sulfides, alloys and their compound materials for the storage of alkaline earth metal ions, including an analysis of the lithium storage mechanism of Co_2_SnO_4_ materials. Dong Sun’s review [39] gives special attention to two tin-based materials, ZnSnO_3_ and Zn_2_SnO_4_, outlining various synthetic strategies, formation mechanisms and morphology and summarizing their various applications. Although some of these excellent previous reviews mention the use of M_2_SnO_4_ for LIBs and SIBs, there is a lack of systematic review articles that address the energy storage mechanisms and material design of these materials. However, M_2_SnO_4_ cathode materials combining alloying and conversion reactions have shown great potential. It is therefore necessary to fill this gap by reviewing the rapid development of M_2_SnO_4_ materials as negative electrodes in the field of rechargeable batteries. In this review, we provide an overview of research on M_2_SnO_4_ as an anode in LIBs and SIBs. We systematically discuss the storage mechanisms, design of materials, and electrochemical properties of M_2_SnO_4_ anode materials in each section (Figure 2). In addition, we summarize and outline the future directions and prospects of M_2_SnO_4_-based anode materials.

## 2. M_2_SnO_4_-Based Anodes for LIBs

### 2.1. Lithium-Ion Storage

Transition metal stannate M_2_SnO_4_ is considered an ideal candidate for LIBs with high theoretical capacity values compared to current commercially available anode electrode materials (graphite) [37,53]. This is mainly since the storage of lithium ions in M_2_SnO_4_ materials (M_2_SnO_4_; M = Zn, Mn, Co, etc.) is carried out by both a conversion reaction mechanism and an alloying–dealloying mechanism. The lithiation mechanism of M_2_SnO_4_ materials is described by a four-stage reaction [23,33,53,54,55,56,57]. As shown in Equation (1), the first step is an irreversible reaction, consisting specifically of the decomposition of M_2_SnO_4_ into M and Sn and the formation of LiO_2_. Next, the second step is the alloying/dealloying reaction as expressed in Equation (2). This reaction allows Sn to bind up to 4.4 Li^+^, resulting in a high capacity, but also causing severe expansion of the Sn volume, which affects cycling stability. Finally, there is a conversion reaction between Sn and LiO_2_/SnO_2_ and M and LiO_2_/MO_2,_ as indicated in Equations (3) and (4). The reversibility of the conversion reaction is a vital factor in maintaining the high capacity of the M_2_SnO_4_-based anode material.
(1)$$M_{2}{\mathrm{SnO}}_{4}+8{\mathrm{Li}}^{+}+{8e}^{-}\to 2M+\mathrm{Sn}+4{\mathrm{LiO}}_{2}\;(M=\mathrm{Zn},\;\mathrm{Mn},\;\mathrm{Co}\;\mathrm{et}\;\mathrm{al})$$
(2)$$Sn+x{\mathrm{Li}}^{+}+xe^{-}↔Li_{x}Sn\;(0<x<4.4)$$
(3)$$Sn+2Li_{2}O↔SnO_{2}+4{\mathrm{Li}}^{+}+4e^{-}$$
(4)$$M+{\mathrm{Li}}_{2}O↔\mathrm{MO}+2{\mathrm{Li}}^{+}+2e^{-}$$

During this lithiation/delithiation cycle, Reactions (3) and (4) form two metals with different redox potentials (metal–lithium alloys). On the one hand, reactions (3) and (4) can provide additional Li_2_O to Sn and Mrespectively, which constitutes a “synergistic effect” and promotes the reversibility of both [33]. On the other hand, the two act as a skeleton for each other to buffer the change in the volume of the material, ensuring the structural integrity of the material and reducing capacity degradation [34]. In addition, Figure 3 shows the CV curves of Mn_2_SnO_4_ and Co_2_SnO_4_, whose responses correspond to the energy storage mechanism described above. It is due to the superiority of this lithium storage mechanism that the development of M_2_SnO_4_ materials as anodes for LIBs is of great significance.

### 2.2. Design of M_2_SnO_4_-Based Anodes for LIBs

#### 2.2.1. Nanostructures

Due to the advantages of nanosizing materials such as increasing the specific surface area and shortening the diffusion path of Li^+^, various structures of electroactive materials are designed and prepared on the nanoscale to improve the rate performance and cycle stability of batteries.

In terms of preparation methods, M_2_SnO_4_-based nanomaterials can be synthesized using a variety of techniques, including hydrothermal/solvothermal methods, solid-phase synthesis and sol–gel methods. Early studies by Irvine et al. [56] demonstrated the successful preparation of nanoscale M_2_SnO_4_ via solid-state synthesis. By comparing the Li^+^ storage behavior of these materials, they found that M_2_SnO_4_ exhibited better reversibility when M = Mn or Zn and poorer capacity reversibility when M = Mg. More recently, hydrothermal methods have emerged as a popular approach for the preparation of M_2_SnO_4_-based nanomaterials due to their advantages in terms of dispersion and ease of control. For instance, pure-phase Mn_2_SnO_4_ [58], Zn_2_SnO_4_ [59] and Co_2_SnO_4_ [23] nanoparticles prepared using this method have been shown to possess excellent electrochemical activity, with initial charge–discharge capacities of 1320, 1900.4 and 1533.1 mAh·g^−1^, respectively.

Wang et al. [23] recently reported a comparison of the properties of Co_2_SnO_4_ nanocrystals prepared using hydrothermal (HT-Co_2_SnO_4_) and high-temperature solid-state (SS-Co_2_SnO_4_) methods. Their results showed significant differences in the electrochemical properties of these materials (Figure 4c). After 50 cycles, the discharge capacity of SS-Co_2_SnO_4_ was 112.8 mAh·g^−1^ with a capacity retention rate of only 12.2%, which is much lower than that of HT-Co_2_SnO_4_ (555.9 mAh·g^−1^ with a capacity retention of 50.3%). SEM and TEM characterization revealed that HT-Co_2_SnO_4_ consisted of uniform spherical nanoparticles with diameters ranging from 80 to 100 nm, while SS-Co_2_SnO_4_ had an irregular shape with smaller particles (about 100 nm) coexisting with larger particles (1–2 μm) (Figure 4a,b). The particle size of the nanocrystals may be the main reason for the difference in electrochemical properties, with smaller nanoparticles obtained by the hydrothermal method exhibiting better electrochemical properties compared to those obtained by the high-temperature solid-state method.

In addition, some unique nanometer-size structures have been designed, such as hollow skeletons [60], hollow nanospheres [54], core–shell nanostructures [61], nanowires [62] and nanoplates [62], which are favorable for buffering volume expansion and maintain structural stability. Using a sol–gel method combined with phase separation, Wang et al. [60] synthesized macroporous Co_2_SnO_4_ with a hollow skeleton (Figure 5a,b) using polyacrylic acid (PAA) as the phase separation agent. As an anode for LIBs, the macroporous Co_2_SnO_4_ exhibited high capacity retention (115.5% at 200 mAh·g^−1^ after 300 cycles) and provided an ultrahigh specific capacity (921.8 mAh·g^−1^ at 1 A·g^−1^). Similarly, Zhang et al. [54] prepared hollow nanospheres (Figure 5c,d) using typical hydrothermal synthesis and heat treatment to serve as an anode for LIBs, which exhibited characteristics such as high multiplicative performance and cycling stability; even at a current density of 1 A·g^−1^, they still could achieve the reversible specific capacity of 442 mAh·g^−1^ after 60 cycles. In contrast to the above preparation methods, one-dimensional nanowires are usually prepared using the vapor phase method. Zn_2_SnO_4_ nanowires (Figure 5e,f)) were synthesized directly on stainless steel substrates by the vapor transport method. Constant current cycling studies of Zn_2_SnO_4_ nanowires at voltages ranging from 0.005 to 3 V and currents of 120 mA·g^−1^ have shown a reversible capacity of 1000 (±5) mAh·g^−1^, which is almost constant for the first 10 cycles and decays to 695 mAh·g^−1^ through 60 cycles thereafter [62]. The design of these special shapes provides additional cushioning space, relieves volume expansion and protects the stability of the structure.

Nanosizing the material and designing it with suitable structures, especially hollow, porous and unique structures, has been shown to effectively improve the bulk expansion of stannate-based materials.

#### 2.2.2. Composited with Carbon Materials

In contrast to nanostructures, which improve material morphology, the incorporation of carbon materials optimizes material composition. By designing a combination of active and carbonaceous materials, the conductivity of the electrode material can be increased and its stability improved by buffering volume expansion. Commonly used carbon sources include glucose [61,63,64], fructose [55], graphene [53,65,66,67,68,69,70,71], carbon nanotubes [72,73] and other organic materials.

Previously, Qi and colleagues [61] synthesized a core–shell nanostructure of Co_2_SnO_4_@C using glucose as the carbon source. The uniform carbon layer (5–10 nm) coated on the surface of the Co_2_SnO_4_ material (Figure 6a) significantly improved the electrochemical performance of the electrode material. The capacity retention rate of Co_2_SnO_4_@C was much higher than that of pure-phase Co_2_SnO_4_ in terms of cycling performance (Figure 6b). EIS test results (Figure 6c) showed that the diameter of the semicircle in the mid-frequency region of Co_2_SnO_4_@C was smaller than that of Co_2_SnO_4_, indicating that it had a smaller charge transfer resistance. Similarly, a core–shell nanorod structure of Zn_2_SnO_4_@C was synthesized using a similar method. The nanorod is composed of continuous stripes with the same orientation and has a lattice spacing of about 0.26 nm and a uniform carbon coating layer of 5–10 nm (Figure 6d) [65]. The electrochemical performance of carbon composite materials was compared with pure-phase materials (Figure 5e,f), and it was found that the combination of carbon materials can effectively improve the stability and charge transfer impedance of the electrode. The presence of this carbon coating layer serves as a buffer base to enhance structural stability on one hand and improves electron conductivity on the other hand to accelerate charge transfer on the surface. Therefore, it can exhibit superior electrochemical performance.

Compared with traditional carbon materials, graphene materials possess excellent electrical conductivity, flexible structure and high specific surface area. In particular, they exhibit greatly improved electrochemical properties when composited with metal oxide anode materials [71,74]. Typically, a Zn_2_SnO_4_/graphene nanohybrid in flake form was prepared using the in situ hydrothermal method. In this anode system, GNSs act as a buffer to mitigate the volume change and as a separator to inhibit the aggregation of nanoparticles, thus improving the cycling stability. Additionally, the addition of GNSs provides a two-dimensional conductive channel for the Zn_2_SnO_4_ nanocrystals, enhancing the rate capability [68]. It is worth mentioning that graphene and M_2_SnO_4_ are combined via an electrostatic mechanism, which generally manifests itself as positively charged M_2_SnO_4_ nanoparticles being uniformly wrapped in negatively charged graphene under the influence of electrostatic forces [71]. Therefore, some method is needed to positively charge the surface of M_2_SnO_4_ to bind to the negatively charged graphene. For example, Co_2_SnO_4_ nanoparticles (Co_2_SnO_4_ NPs) were dispersed in CoCl_2_ solution so that they adsorbed Co^2+^ for positive charge [71]. Similarly, a Co_2_SnO_4_ hollow cube (Co_2_SnO_4_ HC) was modified with aminopropyltriethoxysilane (APTES) so that its surface was functionalized and positively charged (Figure 7a) [67]. As the uniform wrapping of graphene sheets (Figure 7b,c) effectively alleviates the volume change, the electrodes can maintain excellent conductivity throughout the discharge/charging process, so Co_2_SnO_4_ HC@rGO can provide more than 1000 mAh·g^−1^ capacity at 100 mA·g^−1^ after 100 cycles, while Co_2_SnO_4_ NPs@rGO can achieve the large reversible capacity of 1037.9 mAh·g^−1^ after 200 cycles. In addition to the above-listed Co_2_SnO_4_, the graphene-coated Zn_2_SnO_4_ (Figure 7d,e) obtained by electrostatic interaction also exhibits enhanced properties compared to the pure-phase material [69].

In contrast to the direct wrapping of graphene sheets on stannate-based material described above, the work of Rehman et al. [53] designed a bouquet-like nanocomposite structure in which graphene sheets were embedded in nanoparticles containing manganese and tin to achieve a high degree of bonding. The strategy benefits from a unique porous nanostructure in which high electronic pathways provided by graphene sheets are used to enhance electronic conductivity and uniformly distributed nanoparticles accelerate the kinetic reaction with lithium ions. Moreover, the graphene sheet also limits the growth of stannate nanoparticles, reducing the grain size. Due to the unique structure and conductive network of Mn_2_SnO_4_@GS, the Mn_2_SnO_4_@GS anode material exhibits excellent rate and cycling performance. At a current density of 400 mA·g^−1^, a specific capacity of about mAh·g^−1^ can be achieved after 200 cycles, while at a high current density of 2500 mA·g^−1^, a specific capacity of about 455 mAh·g^−1^ can still be provided (Figure 8a).

Two-dimensional graphene sheets and one-dimensional carbon nanotubes are the stars of the carbon materials world. Carbon nanotubes (CNTs) are frequently used as carbon materials compounded with metal oxides due to their superior electrical conductivity, high aspect ratio and large specific surface area [75,76,77]. Three-component Mn_2_SnO_4_@MWCNTs composites with cubic particles and high porosity anchored on carbon nanotubes were synthesized through a facile hydrothermal method [73]. It is clearly demonstrated by the impedance spectra that due to the synergy between Mn_2_SnO_4_ and MWCNTs, the electron transfer resistance in the LIBs is reduced [78], resulting in a much faster charge transfer in Mn_2_SnO_4_@MWCNTs than in bare Mn_2_SnO_4_ (Figure 8b).

In recent years, metal–organic frameworks (MOFs) have been widely investigated as materials that contain both metal and carbon sources with easily controlled structures [79,80,81,82], offering a new approach to the preparation of carbon-coated materials using the pyrolysis of MOFs [83,84]. Shi et al. [33] designed carbon-coated Mn_2_SnO_4_ nanomaterials with a two-dimensional combined micro/nanoscale configuration using a two-step carbonization process based on Mn-based metal–organic frameworks (Mn-MOFs) as a precursor. The Mn-MOFs play a role in providing both the Mn source and the flake-like porous carbon matrix. When comparing the cyclic and rate performance of Mn_2_SnO_4_@C with MnO/SnO_2_@C which has the same elemental composition, it was concluded that the “synergistic effect” in Mn_2_SnO_4_ [36] and the unique 2D structure brought by the MOF precursors contribute to the reversibility of the lithium storage reaction (Figure 8c). In contrast to the two-step method described above, Yue et al. [85] directly synthesized a Zn-Sn binary MOF (ZT-MOF) and then used rapid calcination under a reducing atmosphere to obtain Zn_2_SnO_4_@C/Sn composites. Interestingly, this work explored the effect of calcination time on the electrochemical properties of the material and found that the material calcined for 1 min had the best performance, mainly since metal Sn grew into large particles as the calcination time increased, which transformed the lithium storage process from surface-controlled to diffusion-controlled.

In summary, incorporating carbon materials through various methods to optimize material composition has proven to be an effective approach for mitigating volume expansion and enhancing the electrical conductivity of M_2_SnO_4_ materials. This approach has been widely adopted in the development of M_2_SnO_4_-based anode materials.

#### 2.2.3. Heterogeneous Structures

The coupling between different nanocrystals builds heterogeneous structures at the contact interface, and the resulting Mott–Schottky heterogeneous junction generates an internal electric field at the interface, which improves the electrochemical dynamics [86,87]. However, the rational design and controlled synthesis of nano-heterostructured anode materials with high performance is still a challenge.

The yolk–shell structure is a classical form of a heterogeneous structure. Ju et al. [88] described the preparation of Mn-Sn-O-C composites with yolk–shell heterostructures by simple spray pyrolysis at various temperatures, where the shell and core portions were MnO-Mn_2_SnO_4_-C and Sn-Mn_2_SnO_4_-C, respectively. The Mn-Sn-O-C prepared at 900 °C had good structural stability with a discharge capacity of 784 mAh·g^−1^ over 100 cycles at a current density of 1 A·g^−1^ (Figure 9a–e). Similar to the yolk–shell structure, Tian et al. [89] used a three-step method combining co-precipitation, hydrothermal treatment and carbonization to design multi-yolk–shell SnO_2_/Mn_2_SnO_4_@C nanoboxes. In this report, it is shown that the SnO_2_/Mn_2_SnO_4_ heterostructure on the one hand produced lattice distortions in the internal material to improve thermodynamic stability, and on the other hand promoted reaction kinetics by hindering the coarsening of Sn, inducing a redistribution of electrons between SnO_2_ and Mn_2_SnO_4_ and accelerating the diffusive adsorption of Li^+^ through the internal electric field at the heterogeneous interface. Benefiting from the properties brought about by the heterogeneous structure, SnO_2_/Mn_2_SnO_4_@C nanoboxes as anode materials for LIBs exhibit a large reversible capacity (1293 mAh·g^−1^ at 0.2 A·g^−1^ after 100 cycles) and a stable long cycle performance (more than 549 cycles at 2 A·g^−1^).

Zhuang et al. [34] reported a novel heterogeneous composite material with excellent rate performance and cyclability by constructing a sandwich structure of graphene hollow spheres confined to a Mn_2_SnO_4_/SnO_2_ heterostructure (Mn_2_SnO_4_/SnO_2_@SG) as an anode for LIBs. Microscopically, the Mn_2_SnO_4_ and SnO_2_ nanoparticles form a heterogeneous structure next to each other between the graphene spherical shells; macroscopically, the structure appears to be a sandwiched hollow sphere. The Mn_2_SnO_4_/SnO_2_ Mott–Schottky heterojunction generates a strong electric field at the interface, which greatly facilitates the electronic/ionic transport kinetics and therefore shows excellent rate properties (823.8 mAh·g^−1^ at 5 C) (Figure 9f–h).

The heterogeneous structure is a modification method that involves introducing other materials compositionally to construct special structures. This approach has significantly improved the electrochemical performance of M_2_SnO_4_ anode materials. However, the process of constructing these structures can be complex and challenging.

#### 2.2.4. Heteroatom Doping

Doping heteroatoms into the lattice of a material can generate a large number of active sites and improve electronic conductivity. This approach can be applied to M_2_SnO_4_ nanomaterials and carbon-based materials to enhance their electrochemical properties. Incorporating heteroatoms into these materials holds great potential for improving their performance in various applications [90].

Wang et al. [91] reported for the first time the fabrication of nanocomposites (Co-ZTO-G-C) consisting of ultrafine (3–5 nm) Co-doped Zn_2_SnO_4_ nanoparticles, graphene nanosheets and amorphous carbon layers using a one-step hydrothermal method. Since the radii of Co^2+^ (0.074 nm) and Zn^2+^ (0.074 nm) are so close to each other, the substitution of Co^2+^ for Zn^2+^ in the lattice of Zn_2_SnO_4_ is favorable, and no other byproducts such as cobalt oxide were found by X-ray diffraction analysis, which also proves the successful doping of Co^2+^. In particular, the crystallinity of the Co-ZTO-G-C nanocomposite is significantly higher than that of the other two nanocomposites, indicating that Co doping enhances the crystallinity of the Zn_2_SnO_4_ nanoparticles. The doping of N atoms into carbon-based materials such as N, P, B and S is also a common strategy. Wan et al. [37] doped N atoms into carbon and encapsulated Sn@Mn_2_SnO_4_ nanoparticles in N-doped carbon to be used as anode materials for LIBs. The doping of N atoms enhanced the electrical conductivity of the carbon shell and improved the lithium storage capacity.

Whether through the direct introduction of Co^+^ into nanomaterials to improve their crystallinity or the introduction of N atoms into composite carbon materials to increase active sites and improve electrical conductivity, doping heteroatoms has proven to be a feasible means of modifying M_2_SnO_4_ anode materials.

## 3. M_2_SnO_4_-Based Anodes for SIBs

### 3.1. Sodium Ion Storage

The sodium storage mechanism of the M_2_SnO_4_ material in SIBs involves a multi-step electrochemical redox reaction, which can be expressed by the following Equations (5)–(8) [24,57,92,93]. The initial step is an irreversible decomposition reaction as shown in Equation (5), which specifically includes the decomposition of M_2_SnO_4_ to M and Sn and the formation of NaO. The second step is the alloying/dealloying reaction expressed in Equation (6), where the formation of Na_x_Sn causes the expansion of the volume. The final Equations (7) and (8) represent the conversion reactions of Sn and M.
(5)$$M_{2}{\mathrm{SnO}}_{4}+8{\mathrm{Na}}^{+}+8e^{-}\to 2M+\mathrm{Sn}+4{\mathrm{Na}}_{2}O\;(M=\mathrm{Zn},\;\mathrm{Mn},\;\mathrm{Co}\;\mathrm{et}\;\mathrm{al}.)$$
(6)$$\mathrm{Sn}+x{\mathrm{Na}}^{+}+xe^{-}↔{\mathrm{Na}}_{x}\mathrm{Sn}\;(0\leq x\leq 3.75)$$
(7)$$Sn+2{\mathrm{Na}}_{2}O↔\mathrm{Sn}O_{2}+4{\mathrm{Na}}^{+}+4e^{-}$$
(8)$$M+{\mathrm{Na}}_{2}O↔\mathrm{MO}+2{\mathrm{Na}}^{+}+2e^{-}$$

The introduction of a second metal can enrich redox reactions compared to monometallics [35], and M and MO can be used as “substrates” to attenuate the effect of volume expansion on cycling performance caused by Na_x_Sn generated by alloying reactions during cycling. For example, Huang et al. [94] developed CoMoO_4_@NC using two metals (Co and Mo) as an SIB anode material, which exhibited a long cycle life and maintained a specific capacity of 190 mAh·g^−1^ after 3200 cycles even at a high current density of 1 A·g^−1^.

### 3.2. Design of M_2_SnO_4_-Based Anodes for SIBs

#### 3.2.1. Nanostructures

The use of nanostructures can effectively shorten the diffusion path of Na^+^, facilitate sodiation/desodiation, and reduce the adverse effects of volume expansion during cycling.

Park et al. [57] used the hydrothermal method to directly synthesize high-purity Zn_2_SnO_4_ nanowires and for the first time used them as an anode material for SIBs to investigate their electrochemical properties. In an SIB, Zn_2_SnO_4_ has a reversible capacity of 306 mAh·g^−1^ after 100 cycles at 0.1 C and maintains a high coulombic efficiency of about 99%. Nanowires (NWs) composed of a cubic (Zn_2_SnO_4_) phase have a homogeneous morphology, and the NWs are able to form a network to mitigate the volume change, which improves the cycling performance and extends the battery life. In this study, the working mechanism in Zn_2_SnO_4_ was also investigated by cyclic voltammetry (CV), I–V combined with ex situ XRD, and it was found that the alloying reaction did not proceed completely enough to reach the Na_x_Sn alloy and lead to capacity reduction, while the reversible conversion reaction helped the capacity recovery. The above findings are consistent with previous studies on Sn and SnO_2_ [54,95].

#### 3.2.2. Composited with Carbon Materials

Combining Sn-based materials with carbon materials and designing superior structures can effectively improve the material conductivity as well as the structural stability of electrodes and therefore has received a lot of attention.

Graphene is an excellent carbon material. N. Kalaiselvi’s team [92] designed Mn_2_SnO_4_ nanoporous cubes with graphene and used them for the first time as an anode material for SIBs. To prepare Mn_2_SnO_4_/graphene (MSO/G), the prepared Mn_2_SnO_4_ was sonicated with graphene in an ethanol solution, followed by oven drying and calcination under an argon atmosphere to ensure complete wrapping and adhesion of graphene sheets to the Mn_2_SnO_4_ nanocubes. In the SIB, the MSO/G composite anode exhibited high electrochemical performance at various current rates in terms of cycling capacity (257 mAh·g^−1^ after 100 cycles at 100 mA·g^−1^) and rate capacity (211 mAh·g^−1^ at 500 mA·g^−1^) and reached 106 mA h g^−1^ even after 1000 cycles at a high current density of 1 A·g^−1^. The presence of graphene networks and MSO nanocube voids greatly mitigates the shock caused by volume changes, ensuring that the composite anode material can maintain structural stability even under long cycling and enhancing the sodium storage capacity.

#### 3.2.3. Heterogeneous Structures

According to the related reports, a heterogeneous structure constructed by combining bimetallic and carbon materials introduces defects, disorders and heterogeneous interfaces that can provide effective channels for the diffusion and adsorption of Na^+^, thus promoting the kinetics of the reaction [96,97,98].

Tian et al. designed novel nanoboxes with multiple yolk–shell structures (denoted as SnO_2_/Mn_2_SnO_4_@C) containing heterogeneously structured SnO_2_/Mn_2_SnO_4_ nanoparticles as the yolk and phenolic resin-derived carbon as the shell. The procedure for the preparation of SnO_2_/Mn_2_SnO_4_@C is represented in Figure 8a. Specifically, it includes the following steps: firstly, MnCl_2_, SnCl_4_ and NaOH are used as raw materials to obtain MnSn(OH)_6_ nanoboxes using the co-precipitation method; after that, the phenolic resin is wrapped on the surface of MnSn(OH)_6_ by hydrothermal reaction to obtain PR-MnSn(OH)_6_ nanoboxes; finally, heat treatment is performed to obtain SnO_2_/Mn_2_SnO_4_@C products. During the heat treatment, MnSn(OH)_6_ was transformed into heterostructured SnO_2_/Mn_2_SnO_4_ nuclei and PR was carbonized into carbon shells. As an SIB anode, it provides a large reversible capacity of 203 mA h g^−1^ after 100 cycles. In addition, according to the EIS spectra and fitting results, the SnO_2_/Mn_2_SnO_4_@C anode has a low R_ct_ value (30.55 Ω) and a large D_Na_ coefficient (8.72 × 10^−14^). The excellent electrochemical performance is mainly attributed to the following: the heterogeneous structure of SnO_2_/Mn_2_SnO_4_ effectively mitigates the volume expansion of Sn→Na_x_Sn and improves the reversibility of conversion and alloying, the lattice distortion and the redistribution of charges at the heterogeneous interface accelerate the migration of Na^+^, and the hollow structure of SnO_2_/Mn_2_SnO_4_@C nanoboxes and the phenolic resin-derived carbon shell attenuate the crushing and agglomeration of the material.

#### 3.2.4. Heteroatom Doping

The doping of heteroatoms (N, S, B, P, etc.) into carbon materials can provide more reactive sites and thus effectively improve the electrochemical properties of the materials, so this method of modification at the atomic level has gained great attention [99,100].

Using a typical synthesis method of hydrothermal treatment, carbon encapsulation and high-temperature treatment, Wan et al. [37] prepared the composite Sn@Mn_2_SnO_4_-NC containing Mn_2_SnO_4_ nanoparticles with Sn encapsulated in an N-doped carbon layer. Figure 10a shows synthesis path of Sn@Mn_2_SnO_4_-NC. In an SIB, the introduction of heteroatoms improves the electrical conductivity of the material, facilitates electron/ion transfer and accelerates the reaction kinetics, and the hollow structure combined with the carbon material provides the space needed for volume expansion/contraction and protects the material from crushing. The electrochemical reaction of Sn@Mn_2_SnO_4_-NC in SIBs is mainly controlled by pseudocapacitance according to the CV curves (Figure 10b,c). In addition, the capacitance contribution of SIBs is higher than that of LIBs at the same scan rate, which is mainly attributed to the fact that the radius of Na^+^ is larger than that of Li^+^, which hinders the insertion of ions.

Similarly, Kim et al. [99] reported the synthesis of Zn_2_SnO_4_ nanoparticles with uniformly encapsulated nitrogen-doped carbon layers using dopamine as a single carbon and nitrogen source, and the thickness of the carbon layer was adjusted by tuning the degree of polymerization of dopamine. The presence of the N-doped carbon cladding layer enhances the ionic/electronic conductivity, alleviates the volume expansion during electrode cycling, and prevents the aggregation of nanoparticles and the adverse effects caused by contact with the electrolyte.

## 4. Conclusions and Outlook

In conclusion, we have presented a comprehensive analysis of the development of M_2_SnO_4_-based nanomaterials as anodes for LIBs and SIBs. By incorporating other metallic elements into Sn-based materials to form M_2_SnO_4_ compounds (where M = Zn, Mn, Co, etc.), researchers have enriched the redox reactions involved in energy storage and created metal “M” or metal oxide “MO” buffer substrates. This effectively mitigates the issue of volume expansion during charge/discharge cycles. Table 2 and Table 3 summarizes a range of M_2_SnO_4_-based anode materials, categorized by synthesis techniques, morphological structures and electrochemical performance in LIBs and SIBs.

To improve the electrochemical performance of M_2_SnO_4_ as a battery anode material, researchers have explored various approaches, starting with pure-phase M_2_SnO_4_-based compounds. A primary focus has been on reducing the size of M_2_SnO_4_ materials to the nanoscale and designing various nanostructures (e.g., nanorods, nanowires, nanosheets, hollow nanospheres) to increase specific surface area and shorten Li^+^ and Na^+^ diffusion paths. Nanosized M_2_SnO_4_ materials consistently exhibit superior electrochemical properties compared to their larger counterparts. Three synthesis methods were discussed in the previous section: high-temperature solid phase, hydrothermal/solvothermal and vapor phase. Of these, hydrothermal synthesis is widely used due to its ability to produce high-purity materials with small particle sizes and controllable morphologies.

Subsequently, researchers have combined M_2_SnO_4_ with various carbon carriers to form carbon composites. In these composites, carbon carriers are encapsulated on the M_2_SnO_4_ nanostructure in the form of lamellae or shells. Carbon materials have been compounded with M_2_SnO_4_ using electrostatic forces to combine graphene with M_2_SnO_4_, high-temperature carbonization to encapsulate organic carbon sources (e.g., dopamine, fructose) on the surface of M_2_SnO_4_, and pyrolysis of MOF materials. The incorporation of carbon has a dual effect: it buffers the volume expansion of the active material during charge/discharge cycles and improves the electrical conductivity of the composite. As a result, hybrid M_2_SnO_4_-based materials with carbon exhibit enhanced electrochemical performance as anode materials.

Other researchers have coupled M_2_SnO_4_ with different nanomaterials to create heterogeneous structures with contact interfaces between different nanocrystals. These structures include yolk–shell, sandwich and multi-yolk–shell nanobox structures. The materials chosen for coupling are typically corresponding metal oxides and carbon carriers. The lattice distortion and charge redistribution at the heterogeneous interface between the two nanomaterials facilitate rapid Li^+^ and Na^+^ migration, thus improving reaction kinetics. Additionally, the unique structures created by this strategy can mitigate the crushing and agglomeration of active materials and enhance battery cycling stability.

Although nanosizing enhances the mass-specific capacity of materials, it concurrently diminishes their volumetric loading. Additionally, excessive carbon content may result in inadequate capacity. Modification strategies, such as the formation of heterostructures or doping with foreign elements, often necessitate intricate synthesis procedures. Consequently, selecting suitable modification techniques for distinct materials is of paramount importance. Table 2 and Table 3 provide detailed information on recent developments in M_2_SnO_4_-based anode materials for LIBs and SIBs. These tables summarize the composition, synthesis methods, material morphology and electrochemical properties of these materials.

As performance requirements for M_2_SnO_4_-based anode materials continue to increase, researchers are focusing on reducing particle size and studying the physicochemical properties of these materials in greater depth. However, many conventional experimental tools and characterization methods have significant limitations. Simulation calculations have emerged as an important tool in materials research and are widely used in the design of electrode materials, as well as in the calculation, prediction, validation, optimization and simulation of properties. For material design and preparation, simulation can provide an effective way to avoid unnecessary experiments. For mechanistic studies, it can elucidate the factors influencing electrochemical reactions and reveal microscopic mechanisms at the atomic level. Thus, simulation calculations will be a powerful aid in the study of M_2_SnO_4_-based anode materials.

Volume expansion and its associated problems remain major challenges in the development of M_2_SnO_4_-based materials. To address these issues, researchers are exploring the compounding of stannate materials with various other materials (including carbon carriers, metals and their oxides) and creating unique and refined morphological structures to improve and maintain the high capacity of stannate-based anode materials. Reducing grain size to less than 10 nm could be an effective strategy in terms of material size. In terms of material composition, the heterogeneous interface between different materials and the strength of chemical bonding should be considered. In terms of material morphology, a reasonable layout between the conductive network formed by the carbon material, active material and extra buffer space provided by hollow porous structures is crucial. In terms of heteroatom doping, introducing multiple heteroatoms into a suitable lattice structure could be a direction for future development. For example, while striving for the high weight capacity that nanosizing brings, it is important to also consider the volume capacity required for practical applications.

At the industrial application level, we suggest that using simple preparation methods to obtain M_2_SnO_4_@C composites by combining lower-cost carbon materials with M_2_SnO_4_ is the most promising direction. Although these low-cost carbon materials do not have the exceptional properties of graphene or carbon nanotubes, they can effectively mitigate volume expansion, prevent shedding and aggregation and improve the cycling stability of stannate-based anode materials. Additionally, more attention should be paid to comprehensive performance rather than solely pursuing excellence in one aspect.

M_2_SnO_4_-based materials are already widely used as anode materials in LIBs and SIBs due to their high theoretical capacity. Similarly, potassium-ion, magnesium-ion and calcium-ion batteries have similar reaction mechanisms, and M_2_SnO_4_-based materials could be potential candidates for their anode materials. In addition to these applications, M_2_SnO_4_-based materials may also have potential in emerging energy storage technologies. For example, lithium-ion hybrid capacitors (LICs) are a new type of energy storage device that combines a pre-lithiated anode from an LIB with an electric double-layer capacitor (EDLC)-type cathode. This device combines the high energy density of LIBs with the high power density and long cycle life of EDLCs and is considered to be a highly promising energy storage device. Stannate-based anode materials with high capacity are potential candidates for LIC anodes.

Despite the many challenges in the development of M_2_SnO_4_-based materials, we believe that they have great potential for use in the next generation of high-performance LIBs and SIBs. We hope that this review will be helpful in the development of M_2_SnO_4_-based anode materials.

## Figures and Tables

**Figure 1 molecules-28-05037-f001:**
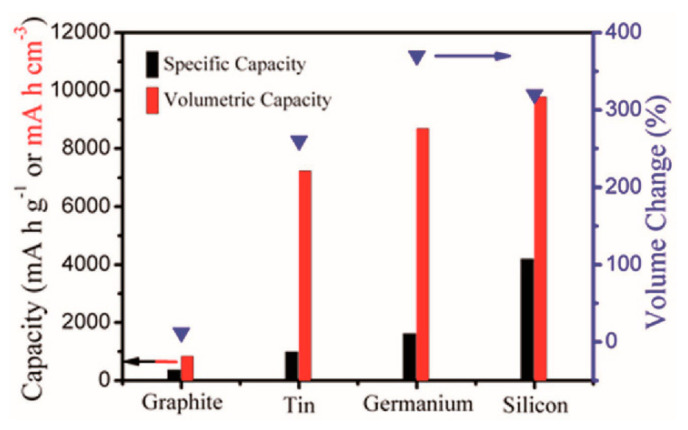
Capacity and volume change comparison of IVA group elements [9]. Copyright 2017, Wiley.

**Figure 2 molecules-28-05037-f002:**
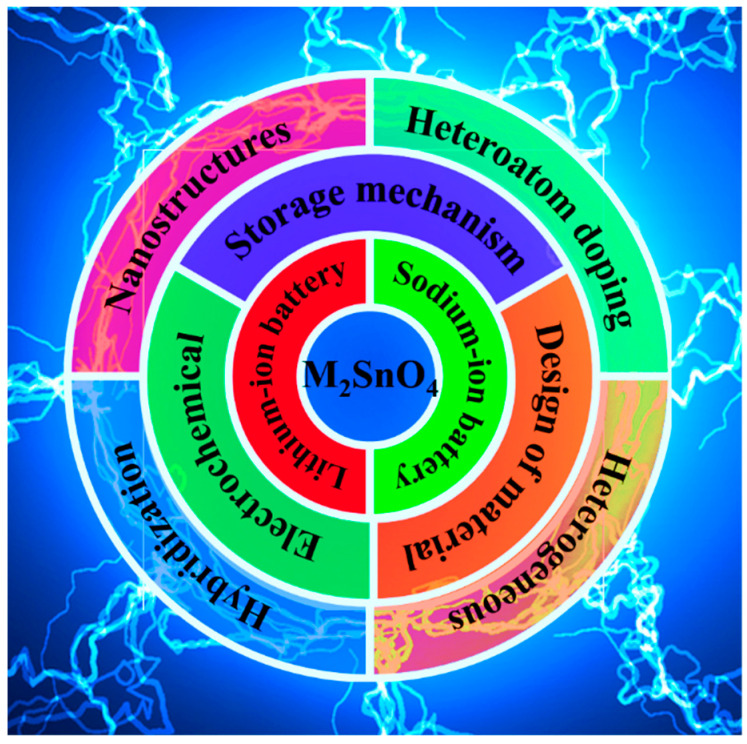
Key elements presented in this overview.

**Figure 3 molecules-28-05037-f003:**
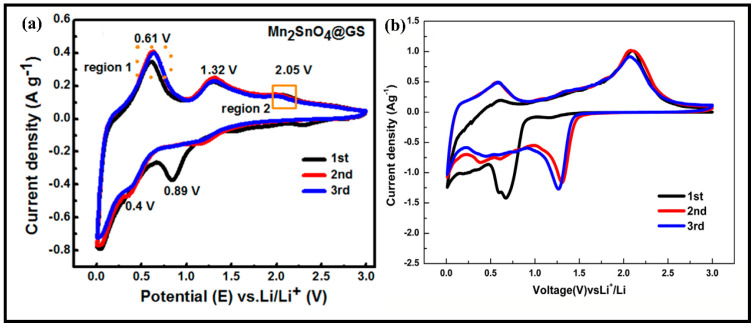
(**a**) First few cyclic voltammetry curves of Mn_2_SnO_4_@GS [53]. Copyright 2018, American Chemical Society. (**b**) First few cyclic voltammetry curves of Co_2_SnO_4_. Copyright 2014, Elsevier.

**Figure 4 molecules-28-05037-f004:**
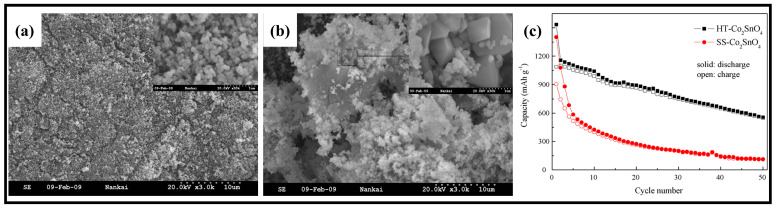
SEM images of the products synthesized via hydrothermal reaction (**a**) and high-temperature solid-state reaction (**b**). SEM images at high magnification are inserted. (**c**) Cycle performance of HT-Co_2_SnO_4_ and SS-Co_2_SnO_4_ at 30 mA·g^−1^ [23]. Copyright 2009, Elsevier.

**Figure 5 molecules-28-05037-f005:**
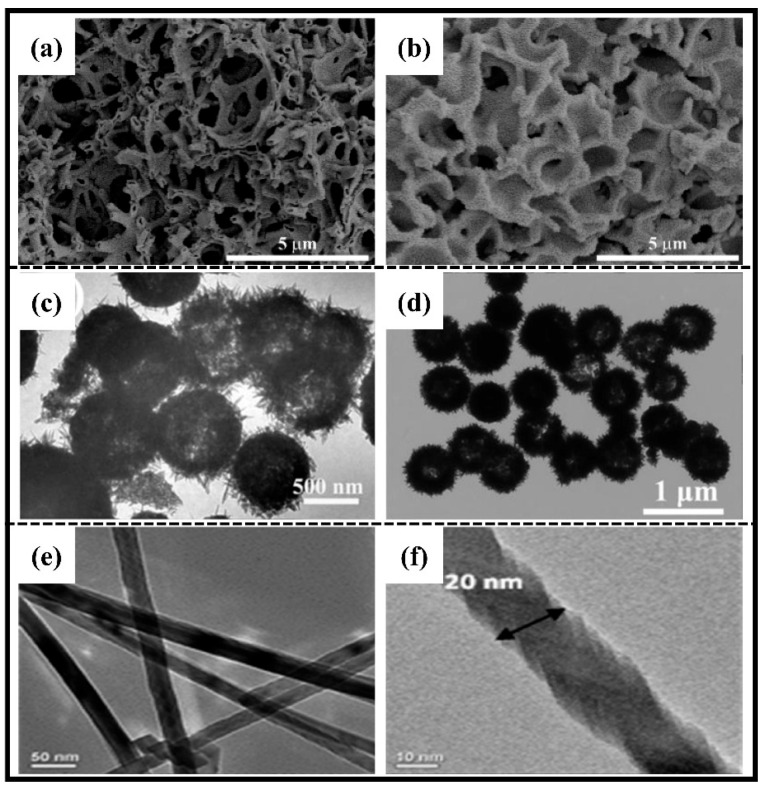
SEM images of (**a**,**b**) Co_2_SnO_4_ hollow skeletons [60]. Copyright 2022, MDPI. TEM images of (**c**,**d**) Zn_2_SnO_4_ hollow nanospheres [54]. Copyright 2014, Royal Society of Chemistry. TEM images of (**e**,**f**) Zn_2_SnO_4_ nanowires [62]. Copyright 2013, American Chemical Society.

**Figure 6 molecules-28-05037-f006:**
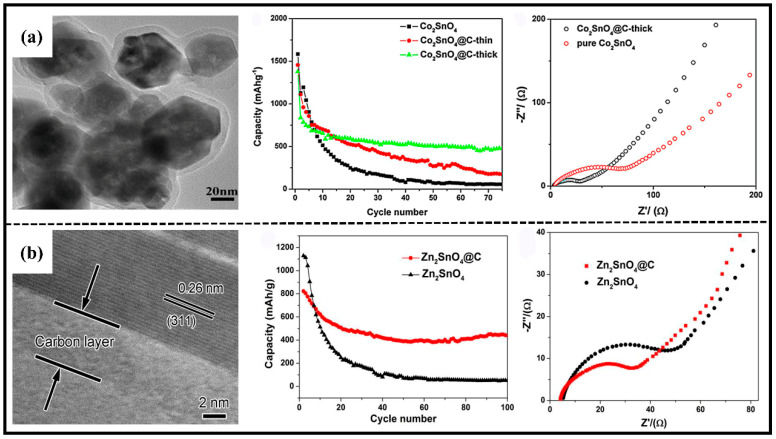
(**a**) TEM images of Co_2_SnO_4_@C-thick nanostructures, cycle performance at a current density of 100 mA·g^−1^ and Nyquist plots [61]. Copyright 2011, Elsevier. (**b**) TEM images of Zn_2_SnO_4_@C core–shell nanorods, cycle performance at a current density of 100 mA·g^−1^ and Nyquist plots [63]. Copyright 2015, Elsevier.

**Figure 7 molecules-28-05037-f007:**
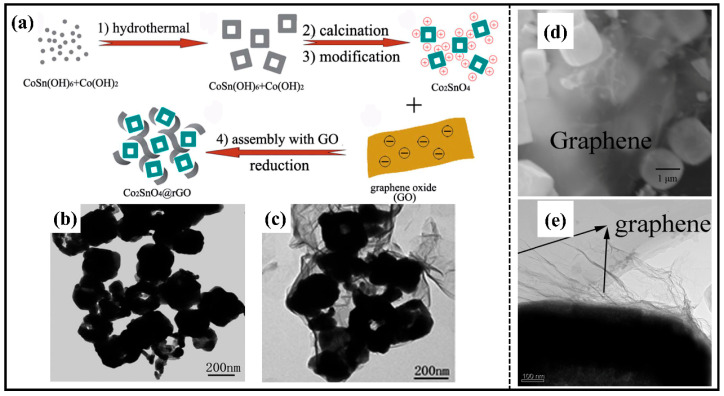
(**a**) Schematic illustration of the formation of Co_2_SnO_4_ HC@rGO; TEM images of (**b**) Co_2_SnO_4_ HC and (**c**) Co_2_SnO_4_ HC@rGO [71]. Copyright 2014, Royal Society of Chemistry. (**d**) SEM and (**e**) TEM images of GN-wrapped Zn_2_SnO_4_ boxes [69]. Copyright 2014, Elsevier.

**Figure 8 molecules-28-05037-f008:**
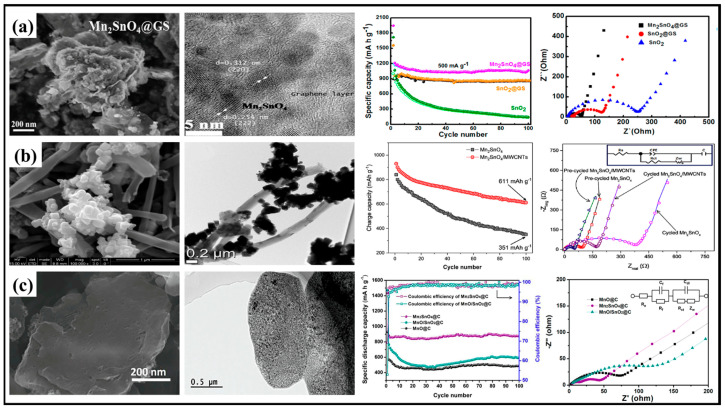
(**a**) SEM and TEM images of Mn_2_SnO_4_@GS, rate capability at various current densities and cycle performance at a current density of 500 mA·g^−1^ [53]. Copyright 2018, American Chemical Society. (**b**) SEM and TEM images of Mn2SnO4@MWCNTs, Nyquist plots and cycle performance at a current density of 100 mA·g^−1^ [73]. Copyright 2020, Elsevier. (**c**) SEM and TEM images of Mn_2_SnO_4_@C, rate capability at various current densities and cycle performance at a current density of 100 mA·g^−1^ [33]. Copyright 2019, Elsevier.

**Figure 9 molecules-28-05037-f009:**
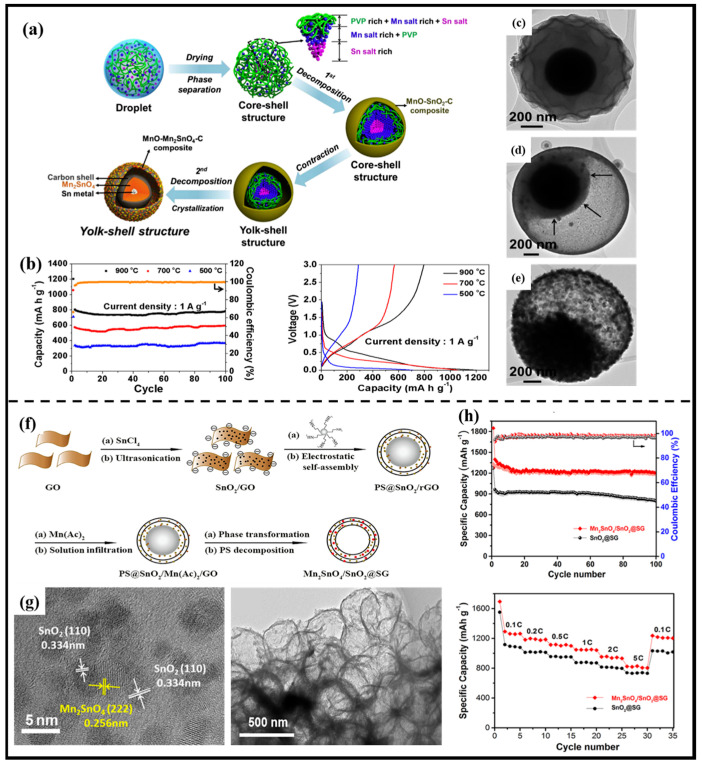
(**a**) Schematic illustration of the fabrication process of multi-yolk–shell SnO_2_/Mn_2_SnO_4_@C; (**b**) cycling performance of powders prepared at various temperatures, initial charge/discharge curves of powders prepared at various temperatures (300 °C/700 °C/900 °C); (**c**–**e**) TEM image of powders prepared at various temperatures [88]. Copyright 2016, Elsevier. (**f**) Schematic illustration of the fabrication process of Mn_2_SnO_4_/SnO_2_@GO; (**g**) SEM and TEM images; (**h**) cycling performance and rate performance at various current densities of Mn_2_SnO_4_/SnO_2_@SG [34]. Copyright 2021, Elsevier.

**Figure 10 molecules-28-05037-f010:**
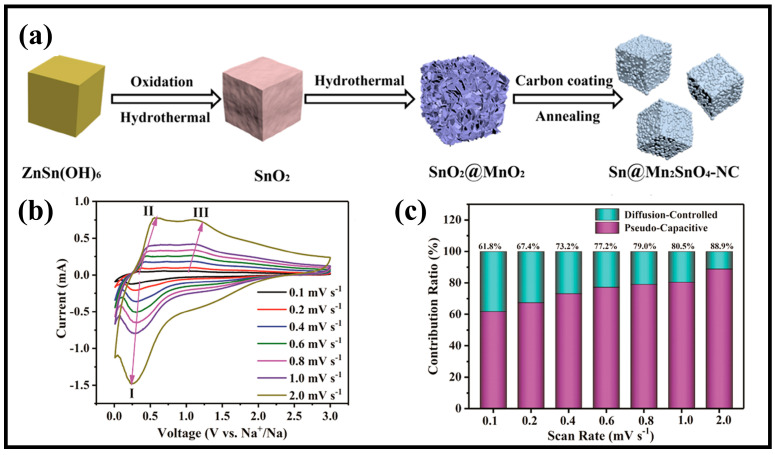
(**a**) Schematic diagram of the synthesis route of Sn@Mn_2_SnO_4_-NC. (**b**) CV curves at various scan rates (I, II, III: specific redox peaks) (**c**) contribution ratios of the pseudocapacitive- and diffusion-controlled capacity at different scan rates for SIBs. Copyright 2021, American Chemical Society.

**Table 1 molecules-28-05037-t001:** Existing review literature on Sn-based materials.

Sr No.	Title of Paper	Reviewed Material	Year	Covers Stannate	Ref.
1	SnO_2_-Based Nanomaterials: Synthesis and Application in Lithium-Ion Batteries	SnO_2_-based nanomaterials	2013	No	[40]
2	SnO_2_-Based Nanomaterials: Synthesis and Application in Lithium-Ion Batteries and Supercapacitors	SnO_2_-based nanomaterials	2015	No	[41]
3	Significant impact of 2D graphene nanosheets on large volume change tin-based anodes in lithium-ion batteries: A review	Sn, SnO_2_, SnS_2_, M_x_SnO_y_ (stannates)	2015	Section	[42]
4	Tin-based anode materials with well-designed architectures for next generation lithium-ion batteries	Sn-based multi-component intermetallics, SnO_2_, SnS_2_	2016	No	[43]
5	Tin-based nanomaterials for electrochemical energy storage	Sn; Sn–M (M-Co, Ni, Cd, Zn, Fe), SnO_2_, SnS_2_, Sn_4_P_3_, SnF_2_	2016	No	[44]
6	Morphological zinc stannate: synthesis, fundamental properties and applications	ZnSnO_3_ and Zn_2_SnO_4_	2017	Only Zn-based	[39]
7	Metallic Sn-Based Anode Materials: Application in High-Performance Lithium-Ion and Sodium-Ion Batteries	Sn; Sn in carbon matrix, Sn alloy	2017	No	[9]
8	Tin-based materials as versatile anodes for alkali (earth)-ion batteries	SnO_2_, SnS_2_, Sn Alloy, M_x_SnO_y_	2018	Section	[38]
9	Advances in Sn-Based Catalysts for Electrochemical CO_2_ Reduction	SnO_2_, SnS_2_, tin alloy and its composites	2019	No	[45]
10	Tin oxide–based anodes for both lithium-ion and sodium-ion batteries	SnO_2_ and its composites	2020	No	[24]
11	Research progress on tin-based anode materials for sodium ion batteries	Sn, SnO_2_, SnSe, SnS and its composites	2020	No	[46]
12	Tin and Tin Compound Materials as Anodes in Lithium-Ion and Sodium-Ion Batteries: A Review	Tin alloy, SnO_2_, SnS_2_ and its composites	2020	No	[47]
13	Challenges and Development of Tin-Based Anode with High Volumetric Capacity for Li-Ion Batteries	Tin alloy and its composites	2020	No	[12]
14	Tin oxide for optoelectronic, photovoltaic and energy storage devices: a review	SnO_2_ and its composites	2021	No	[48]
15	Advances in Synthesis, Properties and Emerging Applications of Tin Sulfides and its Heterostructures	Sn_x_S_y_ and its composites	2021	No	[49]
16	Sn-Based Electrocatalyst Stability: A Crucial Piece to the Puzzle for the Electrochemical CO_2_ Reduction toward Formic Acid	Sn, SnO_2_ and its composites	2021	No	[50]
17	Sn-based nanomaterials: From composition and structural design to their electrochemical performances for Li- and Na-ion batteries	Sn, SnO_2_, SnS, SnS_2_ and its composites	2021	No	[6]
18	Fundamentals and recent progress of Sn-based electrode materials for supercapacitors: A comprehensive review	SnO_2_, SnS_x_ and its composites	2022	No	[51]
19	A review of tin selenide-based electrodes for rechargeable batteries and supercapacitors	SnSe, SnSe_2_ and its composites	2022	No	[52]

**Table 2 molecules-28-05037-t002:** Selected M_2_SnO_4_-based composites and their electrochemical performances for LIBs (ICE: initial coulombic efficiency (%); cycling stability: specific capacity (mAh·g^−1^)/cycle/current density (A·g^−1^)).

Material	Synthesis Method	Morphology	ICE	Cycling Stability	Ref.
Co_2_SnO_4_	hydrothermal reaction	nanoparticles	71	556/50/0.03	[23]
Co_2_SnO_4_@C	hydrothermal and heat treatment	core–shell nanostructures	-	474/75/0.1	[61]
Co_2_SnO_4_@MWCNTs	hydrothermal reaction	3D network	89	898/50/0.05	[72]
Co_2_SnO_4_/Co_3_O_4_	co-precipitation method	spherical and polyhedral	74.4	702/50/0.1	[101]
Co_2_SnO_4_@C	sonochemical and hydrothermal	cubic phase	59	742/30/0.04	[64]
Co_2_SnO_4_ HC@rGO	hydrothermal and heat treatment	hollow cubes	69	1016/100/0.1	[71]
Co_2_SnO_4_@C	co-precipitation process and high-energy ball milling	spherical particles	58	573.8/100/0.2C	[102]
Co_2_SnO_4_/G	hydrothermal	nanoparticles	69	1061/100/0.1	[65]
Co_2_SnO_4_/Co_3_O_4_/Al_2_O_3_/C	co-precipitation	particle-like morphology	-	1170/100/0.1	[103]
Co_2_SnO_4_ NPs@rGO	hydrothermal	nanoparticles	63.4	1037/200/0.2	[67]
SnO_2_/Co_2_SnO_4_@rGOA	sol-gel and heat treatment	cubic phase	70	588/1500/1	[104]
Co_2_SnO_4_/C	sol–gel method combined with phase separation	hollow skeletons	79.5	582/500/1	[60]
Mn_2_SnO_4_	hydrothermal and thermal decomposition	nanoparticles	42	-	[58]
Li_2_MnSnO_4_/C	oxalyl dihydrazide-assisted combustion method	spherical morphology	69	610/100/0.1	[105]
MnO/Mn_2_SnO_4_/C@ Sn/Mn_2_SnO_4_/C	spray pyrolysis	yolk–shell	66	784/100/1	[88]
Mn_2_SnO_4_/Sn/C	hydrothermal and heat treatment	porous cubes	59.6	908/100/0.5	[88]
Mn_2_SnO_4_@GS	hydrothermal and heat treatment	bouquet-like	61	1070/200/0.4	[53]
Mn_2_SnO_4_@rGO	hydrothermal and heat treatment	nanoparticles	-	542/100/0.1	[70]
Mn_2_SnO_4_@C	hydrothermal and heat treatment	flake-like	69.2	986/100/0.1	[33]
Mn_2_SnO_4_@MWCNTs	hydrothermal and heat treatment	cubic and nanotube	72	611/100/0.1C	[73]
SnO_2_/Mn_2_SnO_4_@C	hydrothermal	multi-yolk–shell nanoboxes	50.55	1293/100/0.2	[89]
Sn@ Mn_2_SnO_4_-NC	hydrothermal and heat treatment	cubic frame	73.88	823/600/1	[37]
Mn_2_SnO_4_/C	sol–gel and heat treatment	Dictyophora-shaped hierarchically porous	-	784/500/1	[106]
Mn_2_SnO_4_/SnO_2_@SG	heat treatment	hollow spheres	71.4	1180/100/0.1C	[34]
Zn_2_SnO_4_/C	hydrothermal and carbothermic reduction	nanoparticles	61	563/40/-	[107]
Zn_2_SnO_4_	vapor transport/hydrothermal	nanowires/nanoplates	41	470/50/-	[62]
Zn_2_SnO_4_/G	in situ hydrothermal	layered	54	688/50/0.2	[68]
Mn3O4/Zn_2_SnO_4_	hydrothermal	nanorod/nanoneedle	59.6	529/50/0.5	[108]
Zn_2_SnO_4_	hydrothermal	hollow nanospheres	66.2	602.5/60/0.1	[54]
Zn_2_SnO_4_/G	co-precipitation and alkali etching method	hollow boxes	62	678.2/45/0.3	[69]
Zn_2_SnO_4_@C	hydrothermal and carbonization approach	core–shell nanorods	53.6	495/100/0.1	[63]
Zn_2_SnO_4_/G	hydrothermal	nanoparticles	57.4	492/500/0.5	[66]
Zn_2_SnO_4_–graphene–carbon	hydrothermal	nanoparticles	62.3	461/200/0.2	[91]
Zn_2_SnO_4_	hydrothermal	nanowires	52.5	983/100/0.1	[57]
Zn_2_SnO_4_@C/Sn	calcination	large spheres	92.4	1140/100/0.1	[85]
Co–ZTO–G–C	hydrothermal and heat treatment	-	62.3	695/50/0.1C	[109]
Zn_2_SnO_4_/N-doped carbon composite	hydrothermal and heat treatment	spherical shaped particles	71.2	992.4/100/0.6	[110]
LC@Zn_2_SnO_4_@MnO/C(MOF)	solvothermal method and high-temperature annealing treatment	porous micro/nanostructures	65.9	1185.6/150/0.2	[111]
Zn_2_SnO_4_@V@PC	carbonization	yolk–shell	57.2	438/600/1	[112]

**Table 3 molecules-28-05037-t003:** Selected M_2_SnO_4_-based composites and their electrochemical performances for SIBs (ICE: initial coulombic efficiency (%); cycling stability: specific capacity (mAh·g^−1^)/cycle/current density (A·g^−1^)).

Material	Synthesis Method	Morphology	ICE	Cycling Stability	Ref.
Zn_2_SnO_4_	hydrothermal	nanowires	52.5	306/100/0.1	[57]
Zn_2_SnO_4_/NC	hydrothermal and heat treatment	spherical shaped particles	-	324.4/100	[110]
Mn_2_SnO_4_/G	hydrothermal and heat treatment	nanocubes	45.6	106/1000/1	[92]
SnO_2_/Mn_2_SnO_4_@C	hydrothermal and heat treatment	nanoboxes	65.9	203/100/0.2	[89]
Sn@ Mn_2_SnO_4_-NC	hydrothermal and heat treatment	cubic frame	64	185.8/7000/2	[37]

## Data Availability

Not applicable.

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
