# Peer review of "Stannate-Based Materials as Anodes in Lithium-Ion and Sodium-Ion Batteries: A Review"

_molecules, 2023, doi:10.3390/molecules28135037_

Round 1
Reviewer 1 Report
In this review, the authors studied the stannate-based materials as anode in lithium-ion and sodium-ion batteries. Further, different problems discussed that affect the capacity of the batteries such as size, designing suitable structures, doping with carbon materials and heteroatoms and constructing 20 heterostructures. The manuscript is suggested to be accepted after the following issues are addressed.
1) In the abstract, the authors mentioned general information; the authors should improve and describe all aspects; for example, there is no information about the 2D, MOF etc.
2) In Figure 3, the authors mentioned the SEM images and device capacity with cycles, but the SEM quality is not good. The basic information on the SEM images is not readable.
3) On page 7, the authors mentioned that “By comparing the cycling performance of pure phase Co2SnO4 and Co2SnO4@C, it was found that a uniform and continuous carbon layer buffer matrix has a remarkable improvement in the cycling performance of LIBs (Figure 5a)….” But in First part of Figure 5 is related to the TEM image of materials. The authors should add information related to the TEM.
4) In Figure 5, the authors should add details of each figure; for example, there is no information about why the performance was improved. The authors also add information related to the EIS measurement.
5) The authors also described the other family of materials, such as 2D and MOF materials. The authors should add more sections, such as composite with 2D, TMDs, or MOF.
6) The quality of some figures is not good. The authors should improve the quality of the images.
7) Many spelling, grammatical, units and typo errors are present in this paper, and the authors should double-check and revise them thoroughly.
Many spelling, grammatical, units and typo errors are present in this paper, and the authors should double-check and revise them thoroughly.
Reviewer 2 Report
The review article "Stannate-based Materials as Anode in Lithium-Ion and Sodium-Ion Batteries: A Review" by Zhen-Hai Fu et al. reports the binary stannates and their composites as anode materials for lithium- and sodium-ion batteries. The review summarizes the lithium/sodium storage mechanisms of Stannate-based materials, various modification methods, and prospects for future development directions. The topic is believed to be worthy of study. However, there are still some issues that need to be addressed. Therefore, it is recommended to make minor revisions before acceptance for publication in “Molecules”. Following are the specific comments:
1. In the introduction section, some data or evidence needs to be provided to support your statements.
2. In the introduction section, describe the modification methods such as nanosizing and carbon composites.
3. In the introduction section, a more detailed explanation of the necessity of this work is needed.
4. In sections 2 and 3: The lithium/sodium storage mechanism section needs to be supplemented with actual examples for proof.
5. In the conclusion section: Discuss the limitations of different electrode designs in more detail.
6. In Tables 2 and 3, it is necessary to delete the column of publication date.
7. Please check all figures and tables, modify the content and format of figure captions, and ensure that image copyrights are obtained.
Please carefully check the entire manuscript to avoid errors such as grammar, spelling, subscripts and superscripts.
Round 2
Reviewer 1 Report
accepted in the present form
Minor editing of English language required